# Effectiveness, Adverse Events, and Immune Response Following Double Vaccination with BNT162b2 in Staff at the National Comprehensive Cancer Center (NCCC)

**DOI:** 10.3390/vaccines10040558

**Published:** 2022-04-04

**Authors:** Patrik Palacka, Monika Pol’anová, Alena Svobodová, Jan Žigmond, Katarína Zanchetta, Vlasta Gombárová, Martina Vulganová, Ján Slopovský, Jana Obertová, Ľuboš Drgoňa, Michal Mego, Juraj Pechan

**Affiliations:** 12nd Department of Oncology, Faculty of Medicine, Comenius University, 833 10 Bratislava, Slovakia; jan.slopovsky@nou.sk (J.S.); jana.obertova@nou.sk (J.O.); michal.mego@nou.sk (M.M.); 2National Cancer Institute, 833 10 Bratislava, Slovakia; katarina.zanchetta@nou.sk (K.Z.); lubos.drgona@nou.sk (Ľ.D.); 3Central-European Biotech Institute, Riečna 162/4, 811 02 Bratislava, Slovakia; polanova@vhagy.sk (M.P.); jan.zigmond@gmail.com (J.Ž.); m.vulganova@cbinstitute.eu (M.V.); 4Department of Clinical Biochemistry, National Cancer Institute, 833 10 Bratislava, Slovakia; alena.svobodova@nou.sk; 5Department of Nursing, Faculty of Health Care and Social Work, Trnava University, 917 01 Trnava, Slovakia; 6Hospital Pharmacy, National Cancer Institute, 833 10 Bratislava, Slovakia; vlasta.gombarova@nou.sk; 7Department of Oncohematology, Faculty of Medicine, Comenius University, 833 10 Bratislava, Slovakia; 8Department of Oncosurgery, Faculty of Medicine, Slovak Medical University, 833 10 Bratislava, Slovakia; juraj.pechan@nou.sk

**Keywords:** BNT162b2, effectiveness, adverse events, IgG antibodies, cell-mediated immunity

## Abstract

Vaccination remains the leading strategy against COVID-19 worldwide. BNT162b2 is among the first licensed vaccines with high effectiveness. However, the role of antibody and cell immunity response monitoring after vaccination remains unclear. We conducted a 6-month prospective study involving the employees of NCCC in Slovakia, who were tested for IgG antibody and cell immune responses after double vaccination with BNT162b2. IgG antibodies were detected at 3, 7, and 26 weeks, respectively. At 6 months, blood samples were tested by two different interferon-γ release assays to determine responses to spike protein antigen and nucleocapsid protein antigen of the novel coronavirus. Results were stratified by gender and body mass index (BMI). Statistical significance was set at *p* = 0.05. The medical records of 94 respondents (71 females) were analyzed. The mean age was 40.2 years and the mean BMI was 26.4 kg/m^2^. At 6 months after double vaccination, effectiveness was 97.9%. The side effects of the BNT162b2 vaccine were similar after both doses, with no serious adverse events or new safety signals recorded. The IgG index declined rapidly (*p* < 0.0001), and 42.6% of subjects had positive and 57.4% borderline or negative immune cell response at 6 months (*p* < 0.0001). Both T cell activation and IgG counts were lower in morbidly obese patients when compared to some other BMI categories. This study confirmed an acceptable toxicity profile and the high efficacy of BNT162b2 despite a rapid decline of IgG level and negative cell-mediated immunity response in most subjects. An individualized approach to vaccination could be considered in morbidly obese individuals.

## 1. Introduction

Severe acute respiratory syndrome coronavirus 2 (SARS-CoV-2) belongs to the family *Coronaviridae*, genus *Betacoronavirus,* and subgenus *Sarbecovirus*. Since it is different from both zoonotic coronavirus MERS-CoV and SARS-CoV introduced to humans earlier in the past two decades, it has also been called a novel coronavirus [1]. The first case report of a patient infected with this novel coronavirus came from Wuhan, China [2], and at that time, the World Health Organization (WHO) released the official name of the disease caused by this virus, as coronavirus disease 19 (COVID-19) [3]. On 11 March 2020, the WHO declared COVID-19 as a pandemic [4]. Social distancing and travel restrictions began to come into force along with advice on effective handwashing techniques. The WHO initially recommended the use of masks only by those taking care of infected individuals, but later advised their use even for healthy individuals in community settings [5].

Clinical testing of broad-spectrum antivirals and other drugs for the treatment of COVID-19 infection and trials of the first COVID-19 human mRNA vaccine began simultaneously in March 2020. The safety and immunogenicity of two candidates’ vaccines, BNT162b1 encoding a secreted trimerized SARS-CoV-2 receptor–binding domain and BNT162b2 encoding a membrane-anchored SARS-CoV-2 full-length spike, stabilized in the prefusion conformation, were tested in a phase I study [6]. Both vaccines elicited similar dose-dependent SARS-CoV-2–neutralizing geometric mean titers, but BNT162b2 was associated with a lower incidence and severity of systemic reactions. This supported the selection of BNT162b2 for advancement to a phase II/III trial [7], which showed 95% efficacy in preventing COVID-19 and similar safety to that of other viral vaccines at a median of 2 months. Through 6 months of follow-up, BNT162b2 remained highly efficacious and displayed a favorable safety profile [8].

COVID-19 vaccines induce innate and adaptive immune responses by different mechanisms involving the T cellular response and the B cellular response, which leads to the production of antibodies directed against SARS-CoV-2 antigens [9]. Antibodies against both spike and nucleocapsid antigens are produced to neutralize structural proteins of the virus. Spike (S) is a transmembrane glycoprotein comprised of S1 and S2 regions. S1 region mediates recognition and binding on host cells by interacting with the angiotensin-converting enzyme human 2 (ACE2) receptor, S2 region facilitates fusion and entry of a virus. The majority of S1 is comprised of the receptor-binding domain (RBD), which binds directly to ACE2 and is highly immunogenic [10,11]. The BNT162b1 vaccine induced robust immune responses [12,13], but data were published at short median follow-up. An assessment of the dynamics of antibody response and CD4^+^ or CD8^+^ T cell responses up to 6 months after vaccination with two doses of the BNT162b2 mRNA vaccine was published recently [14].

The objective of this study was to prove the efficacy of vaccination, monitor the adverse events of two BNT162b2 doses, determine IgG plasma levels at 3 weeks followed by weeks 7 and 26, and explore immune cell response at 6 months after double vaccination in a specific population of staff at the National Comprehensive Cancer Center (NCCC) in Slovakia.

## 2. Results

### 2.1. Demography

The medical records of 94 respondents were analyzed. Of these, 71 were females and 23 were males. The mean age was 40.17 ± 12.29 years (23–62 years), mean weight 87.87 ± 11.36 kg (69–112 kg), mean height 1.82 ± 0.066 m (1.71–1.95 m), and consecutive mean body mass index (the weight in kilograms divided by the square of the height in meters, BMI) was 26.35 ± 2.73 kg/m^2^ (21.3–31.74 kg/m^2^). (Table 1). According to the Centers for Disease Control and Prevention definition [15], 4 respondents (4.26%) were underweight (<18.5 kg/m^2^), 48 (51.06%) had normal weight (18.5–24.9 kg/m^2^), 27 (28.72%) were overweight (25.0–29.9 kg/m^2^), 12 (12.77%) were obese (30.0–34.9 kg/m^2^), and 3 (3.19%) had morbid obesity (≥35.0 kg/m^2^). A total of 21 subjects (3 males, 18 females) were non-medical workers and 73 (20 males, 53 females) respondents were medical staff. Of those, 33 were doctors (17 men), 28 female nurses, 4 pharmacists (1 male), 6 medical technicians (1 male), and 2 physiotherapists (1 male).

### 2.2. Efficacy

All subjects were vaccinated with two doses of the BNT162b2 vaccine. The mean time between the first and second vaccine dose application was 27.03 ± 2.55 days (range 18–35 days). At median follow-up 289.98 ± 9.14 days (range 256–305 days), COVID-19 was diagnosed in two subjects (2.15%) after the second vaccination. Therefore, the efficacy of vaccination was 97.87% (2/94).

Twenty-two (23.40%) respondents had COVID-19 before the first vaccination, five (5.3%) before the second vaccination. Fisher’s exact test revealed no statistical significance between genders before and after the first and second vaccinations (*p* = 0.4006, *p* = 1.0000, and *p* = 0.4355). There was also no difference between BMI groups (*p* = 0.2551, *p* = 0.8894, and *p* = 0.3100) Fisher’s exact test did not reveal any significant difference between COVID-19 positivity before and after first vaccination (*p* = 0.0818), before and after the second vaccination (*p* = 1.0000) and before the first and after the second vaccination (*p* = 0.4191). As the number of respondents was too low in some categories for χ^2^ or Fisher’s exact test to be applied meaningfully (and as χ^2^ or Fisher’s exact test compares overall results only), the Wilcoxon–Mann–Whitney test, as a difference in mean number of respondents with COVID-19, was also used to further evaluate the statistical significance between each parameter separately. The difference between before the first and before the second vaccination was statistically significant (*p* = 0.0004) as was the difference between before the first and after the second vaccination (*p* < 0.0001). The difference between before and after the second vaccination was not statistically significant (*p* = 0.2567) (Figure 1), but that finding was probably due to the small sample of COVID-19 positive patients. There was no statistically significant difference between the genders with COVID-19 (*p* = 0.3652, *p* = 0.8210 and *p* = 0.4150, respectively). There was no statistically significant difference between BMI groups with regards to COVID-19 positivity. However, if we assume that the probability of COVID-19 positivity is lower with lower BMI, then there would be statistically significant one-sided *p*-value between normal weight, and morbid obesity (*p* = 0.0281) before the first vaccination; and between normal weight and overweight (*p* = 0.0278) after the second vaccination (Table 2 and Figure 2).

### 2.3. Adverse Events

Adverse events (AEs) were on average 2.9 ± 2.24 per person after the first vaccination and 3.6 ± 2.92 after the second vaccination. However, this difference is not significant (*p* = 0.0699). The overall number of AEs between males and females was not significantly different after either the first or second vaccination (*p* = 0.0716 and *p* = 0.4769, respectively) (Figure 3). There was no statistical difference between BMI groups after the first or second vaccination (Table 3 and Figure 4).

There was a statistically significant difference between males and females in the number of headaches after both the first and second vaccination (*p* = 0.0183 and *p* = 0.0299, respectively) (Table 4). There was also a significant difference between the number of fatigues, fevers, and limb pains after the first and second vaccination (*p* = 0.0083, *p* = 0.0021, and *p* = 0.0033, respectively) (Table 5). Other AEs that respondents mentioned after the first vaccination were eyelash edema, tearing and herpes labialis. Other AEs after the second vaccination were pain in the lumbosacral region and hypertension.

### 2.4. IgG Antibodies

IgG were tested 24.36 ± 4.99, 50.21 ± 12.05 and 187.88 ± 12.79 days after the second dose of the vaccine. Corresponding IgG values were 126.81 index (95% CI 118.55–135.08), 93.55 index (95% CI 83.05–104.06), and 17.68 index (95% CI 11.73–23.64) (Figure 5 and Figure 6). There was a statistically significant difference between the first and second IgG count values (*p* < 0.0001), second and third (*p* < 0.0001), and as to be expected, between the first and third IgG count values (*p* < 0.0001). There was no statistically significant difference between IgG values for males and females for first IgG testing [131.77 index (95% CI 114.75–158.79) vs. 125.21 index (95% CI 115.55–134.86), *p* = 0.5008], second IgG testing [93.76 index (95% CI 72.76–114.76) vs. 93.48 index (95% CI 81.06–105.91), *p* = 0.9822], and third IgG testing [14.16 index (95% CI 7.15–21.17) vs. 18.82 index (95% CI 11.21–26.44), *p* = 0.3634] (Figure 5). There was a statistically significant difference in IgG values between underweight and overweight BMI groups for third IgG testing [6.87 index (95% CI 0.37–13.38) vs. 27.49 index (95% CI 11.14–43.85), *p* = 0.018]; further between subjects with morbid obesity [150.00 index (95% CI 150.00–150.00] and normal weight [124.65 index (95% CI 113.21–136.08, *p* < 0.0001], overweight [138.16 index (95% CI 126.53–149.79), *p* = 0.0463], and obesity [111.87 index (95% CI 76.10–147.65), *p* = 0.0388] for first IgG testing; between individuals with morbid obesity [99.25 index (95% CI 45.11–150.00] and normal weight [93.42 index (95% CI 78.66–108.18, *p* < 0.0001] for second IgG testing; and finally between subjects with morbid obesity [7.63 index (95% CI 3.21–12.05] and normal weight [13.50 index (95% CI 9.15–17.85, *p* = 0.0198], and overweight [27.49 index (95% CI 11.14–43.85, *p* = 0.0199] for third IgG testing (Table 6 and Figure 6).

### 2.5. Cell Immunity

The mean time from the second vaccination to QuantiFERON collection was 157.24 ± 101.33 days (males 195.39 ± 109.29 days and females 144.89 ± 96.20 days). The mean values of QuantiFERON Ag1 (CD4+), QuantiFERON Ag2 (CD4+ and CD8+), and QuantiFERON Ag3 (CD4+ and CD16+) were 0.28 ± 0.86, 0.33 ± 1.02, and 0.66 ± 1.67. Sixty respondents had overall interferon-γ (IFN-γ) >10 IU/mL. Those with lower value had a mean of 6.96 ± 2.4 IU/mL. The difference between males and females was not significant for any of these tests, i.e., *p* = 0.4131, *p* = 0.6799, *p* = 0.8703, and *p* = 0.6461 (for those with overall IFN-γ > 10 IU/mL) (Figure 7). However, there were significant differences between BMI groups for QuantiFERON Ag1 and QuantiFERON Ag3 tests as shown in Table 7 and Figure 8.

A total of 48 (51.06%) respondents had negative immunity cell response, 6 (6.38%) border line, and 40 (42.55%) positive. The corresponding results were 34.78%, 8.70%, and 56.52% for males; 56.34%, 5.63%, and 38.03% for females. The difference between gender was not significant (*p* = 0.1500). Borderline results were only present in normal weight and overweight individuals while other subjects had either positive or negative immune cell response. The difference between BMI groups was not significant (*p* = 0.9171) (Table 8).

## 3. Discussion

In this study, we confirmed high vaccine efficacy as defined by the number of COVID-19 positive subjects at or after day 7 following the second dose of BNT162b2 (97.87%). Moreover, we recorded significantly more patients with COVID-19 before the first dose of vaccine compared to the time after double vaccination with BNT162b2. Regarding gender, there were no significant differences, however, COVID-19 was significantly more often diagnosed before vaccination in subjects with morbid obesity than normal weight and after the second vaccination in overweight vs. normal weight.

In a population of 94 NCCC staff, two cases of COVID-19 positivity were recorded despite double vaccination. The first case was a 29-year-old female nurse who tested positive for COVID-19 Delta variant 191 days after the second vaccine dose application by RT-PCR (CT E gene of 15.9). Seven days before positive testing, the measured IgG antibodies were 14.95 index. Despite a high viral load (tonsil swab) and low IgG antibody plasma level, the course of the disease was mild, including a fever up to 38.1 °C, chills, sore throat, headache, and clogged nasal cavities. All symptoms and signs disappeared within 2 days after symptomatic treatment (paracetamol, naproxen, and calcium syrup). After infection, the level of virus-neutralizing antibodies was not investigated, but cell-mediated immunity was determined within this study 73 days after RT-PCR positive testing and concluded as borderline.

The second case was a 49-year-old male physician who tested positive for COVID-19 Delta variant 256 days after the second vaccine dose administration by RT-PCR (CT E gene of 30.0). The IgG antibodies measured 173 days after the second vaccine dose were 7.36 index. Cell-mediated immunity determined 244 days after double vaccination was concluded as negative. In this person, the course of the disease was mild to moderate, including a fever up to 39.0 °C, generalized arthralgia and myalgia, headache, and dry cough. Symptoms and signs disappeared over 4 days. On day 3 after positive testing, this patient was given a monoclonal antibody *bamlanivimab* 700 mg in a single, 1-h, intravenous infusion. The virus neutralizing antibodies were not investigated after infection.

The high effectiveness of BNT162b2 is consistent with 6 months of follow-up among the participants aged 16 years and older in a large placebo-controlled phase III trial [8], in which a vaccine efficacy of 86% to 100% was seen across countries and populations of differing age, gender, ethnic group, and risk factors (such as high BMI) for COVID-19 among individuals without evidence of previous infection with the novel coronavirus. Vaccine efficacy against severe disease was 96.7%. Healthcare workers are a population worthy of special consideration due to recognized high exposure and their potential role in the transmission of SARS-CoV-2. In the SIREN study [16], the effectiveness of the BNT162b2 vaccine was 85% seven days after two doses in a cohort of healthcare workers undergoing regular asymptomatic testing. In both previously mentioned trials, the same definition of BNT162b2 vaccine effectiveness was used as in our study.

In this study, the overall incidence of adverse events was non-significantly higher after the second dose of vaccine compared to the first dose. No difference in adverse events between male and female and different BMI groups was observed. Among the most frequent side effects of the first vaccine dose were injection site pain (70.2%), limb pain (56.4%), and fatigue (36.2%). The second dose was mostly accompanied by injection site pain (62.8%), fatigue (56.4%), and limb pain (34.0%). The overall incidence of headache after the first and second dose of a vaccine was 16.0% and 26.5%, respectively. No subject within this study required a leave of absence due to adverse events associated with the vaccination. Headache was present in significantly more females than males after both vaccinations. While the incidence of fatigue and fever was significantly higher after the second dose of the vaccine, the incidence of limb pain events declined. No serious adverse events of grade 3/4 were recorded.

Previously, we presented data on a prospective study [17] with 413 employees (89 men; 306 healthcare professionals) at NCCC with median age of 47 years (range 19–79 years) aimed at exploring the incidence and severity of adverse events after administration of the first dose of BNT162b2. At a median follow-up of 4 weeks, the median number of mRNA vaccine adverse events was significantly higher in females compared to males and healthcare professionals compared to non-healthcare workers. All side effects were mild and no new safety signals were recognized.

Kadali et al. published a randomized, cross-sectional study [18] performed to explore the adverse events of the BNT162b2 vaccine using an online questionnaire to gather responses from 803 healthcare workers. While the results of this study did differentiate between populations based on age, gender, and level of education, they referred only to adverse events following vaccination without specifying with which dose they were associated. The most common symptoms after vaccination were a sore arm or pain at the injection site (88.04%), fatigue (58.90%), and headache (44.83%). The incidence of both injection site pain and fatigue was comparable with our study, while headache was more often present in this cross-sectional trial.

In our study, we present our results stratified by gender and BMI; moreover, we strictly differentiate between the first and second doses of the vaccine in terms of adverse events. The most common side effects of BNT162b2 were generalized weakness/fatigue and sore arm/pain with similar incidences for each.

In the current study, IgG counts were measured at 3, 7, and 26 weeks. Significant declines in IgG plasma levels were exposed between weeks 3 and 7, as well as weeks 7 and 26. No differences in IgG between males and females were discovered at week 3, week 7, nor week 26. However, there were significantly higher IgG counts in overweight compared to underweight and morbid obesity vs. normal weight, overweight, and obesity BMI groups at week 3. At week 7, IgG counts were significantly higher in morbid obesity compared to normal weight BMI groups, while at week 26, IgG values in morbidly obese subjects were significantly lower than in normal weight and overweight individuals. These data must be interpreted with caution due to the small number of subjects mostly in morbidly obese and underweight groups. Briefly, our study explored the differences in IgG counts in subjects of varying BMI status at set intervals and confirmed a robust increment followed by a rapid decline of plasma IgG over 6 months after the second dose of BNT162b2.

In a phase I/II study of mRNA vaccine BNT162B1 [12], only small increases in novel coronavirus–neutralizing geometric mean titers were observed 21 days after the first dose in adults. However, 7 and 14 days after the second dose, substantially greater serum neutralizing geometric mean titers were achieved compared to human convalescent sera. In another study [13], the antibody responses elicited by BNT162b1 were very similar to those observed in the trial mentioned previously. In summary, BNT162b1 induced robust IgG plasma levels. However, neither analysis assessed immune responses beyond 2 weeks after the second dose of the vaccine.

An assessment of the changes in antibody response up to 6 months after two doses of the BNT162b2 vaccine was recently published by Naaber et al. [14]. Their findings showed strong Spike receptor binding domain antibody responses 1 week after the second dose, followed by a significant decline at 3 and 6 months afterwards. In older individuals, a weaker antibody response correlating with fewer side effects at the time of vaccinations was recognized.

At a mean time of 157 days from double vaccination to immune cell response testing, mean values of QuantiFERON Ag1 activating CD4+ T cells, QuantiFERON Ag2 activating both CD4+ T cells and CD8+ Natural killer cells, and QuantiFERON Ag3 activating CD4+, CD8+, and CD16+ (both T and B cells) were higher than the cut-off (0.15–0.20 IU/mL). Overall interferon-γ (IFN-γ) >10 IU/mL was shown in 63.8% (60/94) of subjects. In individuals with morbid obesity, both QuantiFERON Ag1 and QuantiFERON Ag3 values, reflecting the activation of T cells, were significantly lower when compared to normal weight and overweight subgroups. However, this difference might have been affected by both a small number of individuals and a shorter time interval between the second dose of vaccination and cell-mediated immunity testing in morbidly obese subjects, which was caused by vis major. Overall, positive immune cell response was evident in significantly fewer subjects than borderline and negative responses.

Two doses of BNT162b1 elicited robust CD4^+^ and CD8^+^ T cell responses. The majority of tested individuals had T helper type 1 (T_H_1)-skewed T cell immune responses with RBD-specific CD8^+^ and CD4^+^ T cell expansion. IFN-γ was produced by a large fraction of RBD-specific T cells (CD8^+^ and CD4^+^) [13]. Three months after vaccination with two doses of BNT162b2, 87% of vaccinated individuals developed either CD4+ or CD8+ T cell responses. In addition, CD4+ T cell response was decreased among subjects with elevated senescent CD8+ T cells re-expressing CD45RA (TEMRA) cells [14].

In conclusion, this prospective study confirmed the high efficacy of the BNT162b2 vaccine (more than 97%) in a specific population of National Comprehensive Cancer Center staff, despite a negative or borderline immune cell response in the majority of individuals and a rapid decline of IgG antibody response 6 months post double vaccination. The decline in antibody responses was more marked in morbidly obese individuals who also had a lower activation of T cells. However, the obtained data must be interpreted with caution due to the small number of subjects in this single-center study, which are its major limitations, and therefore, they must be confirmed in a larger population. Based on this study’s results, we believe that consideration should be given to monitoring immune cells and IgG antibody counts in order to optimize the timing of the administration of booster doses of the BNT162b2 vaccine.

## 4. Methods

### 4.1. Study Design and Inclusion/Exclusion Criteria

We conducted a prospective, non-randomized, single-center observational study to explore specified outcomes. The primary objective was to determine the efficacy of vaccination defined as no COVID-19 disease confirmed by real-time reverse transcriptase polymerase chain reaction (RT-PCR), loop-mediated isothermal amplification (LAMP) test, rapid PCR test, or rapid antigen test ≤7 days after the second vaccine dose. The secondary objectives were to study IgG antibodies dynamics over time and cell immunity at 6 months after the second vaccine dose. Inclusion criteria were as follows: Current workers at the National Comprehensive Cancer Center (NCCC) in Slovakia who had received two doses of the BNT162b2 vaccine, age ≥ 18 years, and a time period of at least 7 days after the second vaccine dose application. Subjects who did not meet these inclusion criteria were excluded from the study. After obtaining the approval of the Ethical Committee at the NCCC in Bratislava, Slovakia (protocol code: COVID-SK001), all data were entered by investigators into electronic data files and their accuracy was validated for each subject by an independent investigator.

### 4.2. Vaccine Handling and Vaccination

All subjects were vaccinated at the NCCC in Bratislava (Slovakia) between December 2000 and March 2021 in accordance with the WMA Declaration of Helsinki [19] and the guidelines for good clinical practice [20]. The process of vaccine handling was following: Multidose frozen vials of BNT162b2 were transferred to an environment of 2–8 °C, then thawed for 30 min at temperatures of up to 30 °C. The thawed vaccine was diluted with 1.8 mL sodium chloride 9 mg/mL (0.9%) solution for injection. Diluted vaccines were stored at 2–30 °C and used within 6 h. After dilution, the vial contained 2.25 mL from which 6 doses of 0.3 mL were extracted. Each vaccine was administered intramuscularly into the deltoid muscle of the upper arm [21]. These methods all conform to the manufacturer’s recommendations and the product license.

### 4.3. IgG Antibody Measurement

Blood samples from 94 subjects were tested by the Atellica^®^ IM SARS-CoV-2 IgG (sCOVG), a fully automated 2-step sandwich immunoassay based on indirect chemiluminescent technology for the qualitative and quantitative detection of IgG neutralizing antibodies to the RBD of the S1 spike antigen of SARS-CoV-2 [22]. Results were reported in index values as nonreactive <1.00 index (these samples were considered negative for IgG antibodies) or reactive ≥1.00 index (these samples were considered positive for IgG antibodies). The analytical measuring interval was 0.50–150.00 index.

### 4.4. Cell Immunity

To validate obtained data, blood samples from 94 subjects were tested by two different interferon- γ release assays (IGRAs): QuantiFERON^®^ SARS-CoV-2 ELISA consisting of three antigen tubes (Ag1, Ag2, and Ag3 = Ag1 + Ag2) [23] and CoviFeron SARS-CoV-2 sets consisting of three antigen tubes: Original SP antigen tube to assess IFN-γ responses to SARS CoV-2 Spike Protein (SP) antigen derived from SARS CoV-2 and 20I/501YV1(UK) variant; variant SP antigen tube to assess IFN-γ responses to SARS CoV-2 Spike Protein (SP) antigen derived from SARS CoV-2 and 20H/501.V2/South Africa) and 20H/501.V3 (Brazil) variants; NP antigen tube to assess IFN-γ responses to SARS CoV-2 Nucleocapsid Protein (NP) antigen [24]. The CoviFeron SARS-CoV-2 set was used for data verification. The overall interferon-γ was determined by enzyme-linked immunosorbent assay (ELISA) [22,23].

### 4.5. Statistical Analysis

Data were summarized by frequency for categorical variables and by mean ± standard deviation and range for continuous variables. Results were stratified by gender and BMI group as necessary. *p*-values for categorical variables were calculated using χ^2^ or Fisher’s exact test. *p*-values for continuous variables were calculated using the T-test for normally distributed values and the Wilcoxon–Mann–Whitney test was used for non-normally distributed values. The value of statistical significance was set to 0.05. Data were analyzed using software SAS 9.4 TS Level 1M7 X64_10PRO platform, North Carolina, USA [25].

## Figures and Tables

**Figure 1 vaccines-10-00558-f001:**
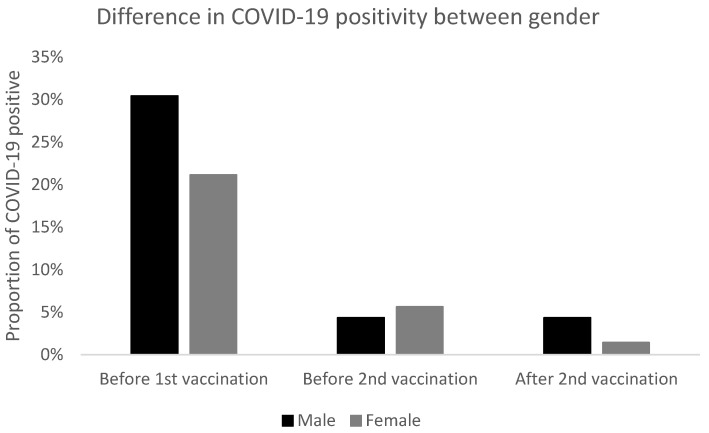
Difference in COVID-19 positivity (males and females) before first and second vaccination and after second vaccination.

**Figure 2 vaccines-10-00558-f002:**
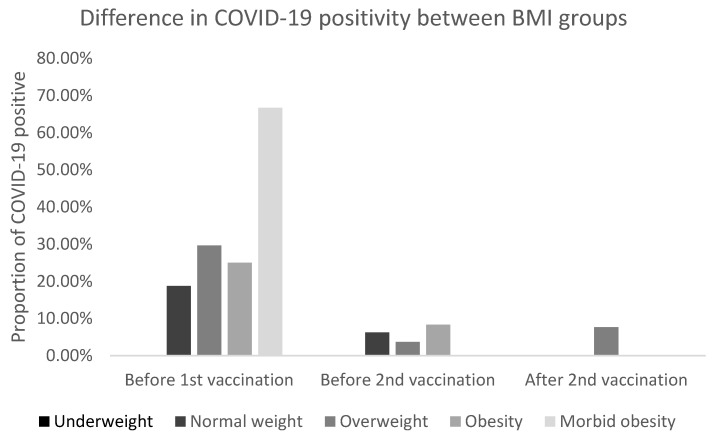
Difference in COVID-19 positivity between body mass index (BMI) groups.

**Figure 3 vaccines-10-00558-f003:**
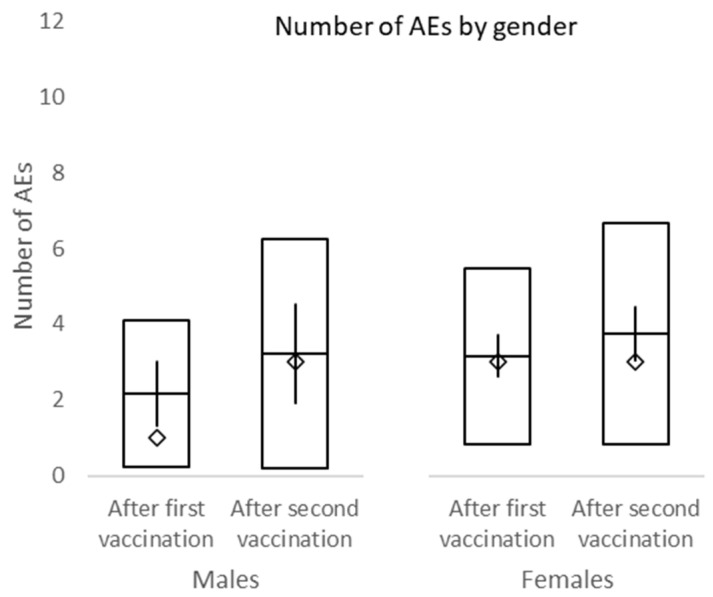
Number of adverse events by gender. □ mean ± standard deviation; ◊ median; │minimum−maximum.

**Figure 4 vaccines-10-00558-f004:**
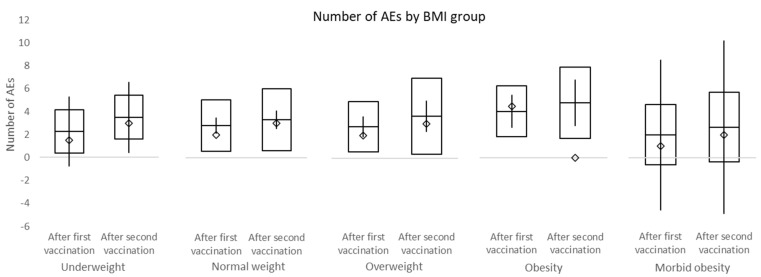
Number of adverse events by body mass index (BMI) groups. □ mean ± standard deviation; ◊ median; │minimum−maximum.

**Figure 5 vaccines-10-00558-f005:**
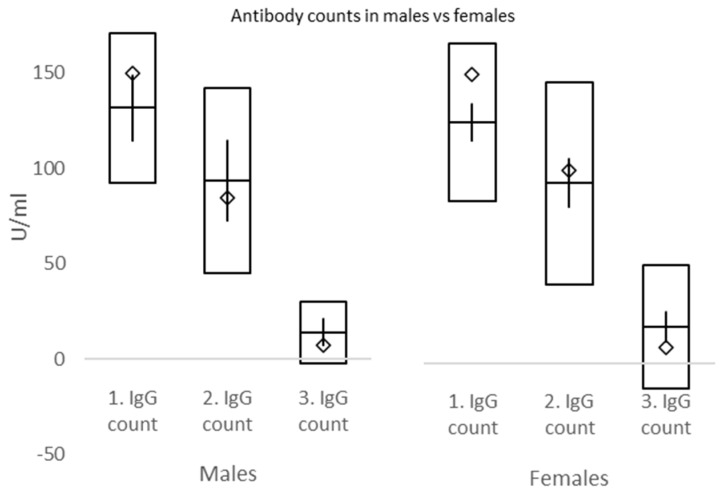
Difference of IgG counts between gender (71 females and 23 males for each IgG measurement) □ mean ± standard deviation; ◊ median; │minimum−maximum.

**Figure 6 vaccines-10-00558-f006:**
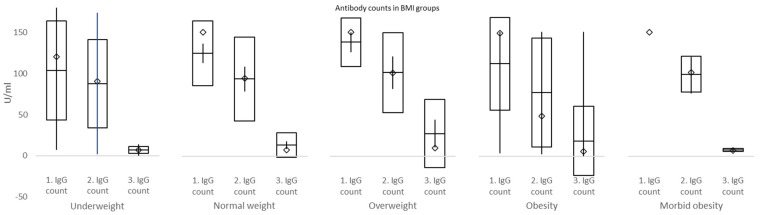
Difference of IgG counts between body mass index (BMI) groups (in underweight, normal weight, overweight, obesity, and morbid obesity subgroups were 4, 48, 27, 12, and 3 subjects, respectively) □ mean ± standard deviation; ◊ median; │minimum−maximum.

**Figure 7 vaccines-10-00558-f007:**
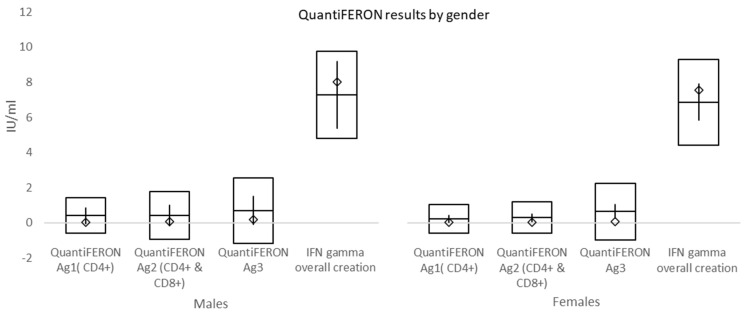
QuantiFERON results by gender. Mean values of QuantiFERON Ag1 activating CD4+ T cells, QuantiFERON Ag2 activating both CD4+ T cells and CD8+ Natural killer cells, and QuantiFERON Ag3 activating CD4+, CD8+, and CD16+ (both T and B cells) were higher than cut-off (0.15–0.20 IU/mL). Overall interferon-γ (IFN-γ) >10 IU/mL was shown in 63.8% of subjects. There was no significant difference between males and females for any of these tests. □ mean ± standard deviation; ◊ median; │minimum − maximum.

**Figure 8 vaccines-10-00558-f008:**
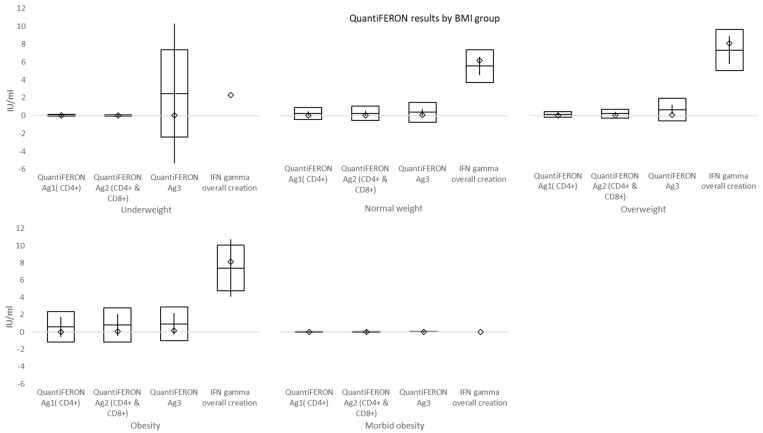
QuantiFERON results by body mass index (BMI) groups. Mean values of QuantiFERON Ag1 activating CD4+ T cells, QuantiFERON Ag2 activating both CD4+ T cells and CD8+ Natural killer cells, and QuantiFERON Ag3 activating CD4+, CD8+, and CD16+ (both T and B cells) were higher than cut-off (0.15–0.20 IU/mL). Overall interferon-γ (IFN-γ) >10 IU/mL was shown in 63.8% of subjects. T cells were significantly less activated in morbid obesity compared to both normal weight and overweight individuals. □ mean ± standard deviation; ◊ median; │minimum − maximum.

**Table 1 vaccines-10-00558-t001:** Demography of respondents. BMI: body mass index; N: number of subjects; SD: standard deviation.

	Male (N = 23)		Female (N = 71)		Total (N = 94)	
	Mean	SD	Mean	SD	Mean	SD
Age	40.17	12.29	47.49	11.45	45.70	12.02
BMI	26.35	2.73	25.22	5.79	25.50	5.22
Weight	87.87	11.36	70.38	15.29	74.66	16.24
Height	1.82	0.07	1.67	0.06	1.71	0.09

**Table 2 vaccines-10-00558-t002:** Two-sided *p*-values for difference in COVID-19 positivity between body mass index (BMI) groups. * Significant if one-sided test was assumed.

		Underweight	Normal Weight	Overweight	Obesity	Morbid Obesity
Before first vaccination	Underweight		0.3592	0.2284	0.3248	0.1175
	Normal weight			0.2867	0.6407	0.0561 *
	Overweight				0.7847	0.2198
	Obesity					0.2169
	Morbid obesity					
Before second vaccination	Underweight		0.6401	0.7728	0.6650	1.0000
	Normal weight			0.6500	0.8140	0.6944
	Overweight				0.5773	0.8241
	Obesity					0.7389
	Morbid obesity					
After second vaccination	Underweight		1.0000	0.6213	1.0000	1.0000
	Normal weight			0.0555 *	1.0000	1.0000
	Overweight				0.3506	0.6835
	Obesity					1.0000
	Morbid obesity					

**Table 3 vaccines-10-00558-t003:** Two-sided *p*-values for difference in adverse events between body mass index (BMI) groups.

		Underweight	Normal Weight	Overweight	Obesity	Morbid Obesity
First vaccination	Underweight		0.6296	0.6776	0.1647	0.8888
	Normal weight			0.8942	0.0847	0.5494
	Overweight				0.0894	0.5921
	Obesity					0.1844
	Morbid obesity					
Second vaccination	Underweight		0.8924	0.9235	0.4409	0.6729
	Normal weight			0.6173	0.0953	0.6902
	Overweight				0.3105	0.6234
	Obesity					0.1844
	Morbid obesity					0.3012

**Table 4 vaccines-10-00558-t004:** Incidence of adverse events by gender. * Other adverse events that respondents mentioned after first vaccination were eyelash edema, tearing, and herpes labialis; after second vaccination: pain in the lumbosacral region and hypertension. These were all >1%. * The significant difference.

	Adverse Events after 1st Vaccine Dose				Adverse Events after 2nd Vaccine Dose			
	All	Females	Males	*p*-Value	All	Females	Males	*p*-Value
Headache	16.0%	21.1%	0.0%	0.0183 *	26.6%	32.4%	8.7%	0.0299 *
Muscle pain	17.0%	21.1%	4.4%	0.1067	25.5%	26.8%	21.7%	0.7856
Joint pain	14.9%	16.9%	8.7%	0.5052	22.3%	22.5%	21.7%	1.0000
Injection site pain	70.2%	67.6%	78.3%	0.4349	62.8%	62.0%	65.2%	1.0000
Fatigue	36.2%	38.0%	30.4%	0.6207	56.4%	57.8%	52.2%	0.8093
Fever	5.3%	5.6%	4.4%	1.0000	21.3%	21.1%	21.7%	1.0000
Injection site swelling	12.8%	12.7%	13.0%	1.0000	16.0%	16.9%	13.0%	1.0000
Nausea	4.3%	5.6%	0.0%	0.5689	9.6%	8.5%	13.0%	0.6837
Injection site redness	12.8%	15.5%	4.4%	0.2819	9.6%	9.8%	8.7%	1.0000
Lymphatic nodes enlargement	10.6%	11.3%	8.7%	1.0000	13.8%	14.1%	13.0%	1.0000
Insomnia	4.3%	4.2%	4.4%	1.0000	9.6%	11.3%	4.4%	0.4452
Limb pain	56.4%	60.6%	43.5%	0.2260	34.0%	35.2%	30.4%	0.8021
Injection site itching	6.4%	8.5%	0.0%	0.3303	4.3%	4.2%	4.4%	1.0000
Lethargy	13.8%	15.5%	8.7%	0.5096	20.2%	19.7%	21.7%	1.0000
Soreness	1.1%	1.4%	0.0%	1.0000	3.2%	2.8%	4.4%	1.0000
Other *	2.1%	2.8%	0.0%	1.0000	2.1%	2.8%	0.0%	1.0000

**Table 5 vaccines-10-00558-t005:** Incidence of adverse events by body mass index (BMI) groups. * Other adverse events that respondents mentioned after first vaccination were eyelash edema, tearing, and herpes labialis; after second vaccination: pain in the lumbosacral region and hypertension. These were all >1%.

	After 1st Vaccination					After 2nd Vaccination				
	Under-weight	Normal Weight	Over- weight	Obesity	Morbid Obesity	Under-weight	Normal Weight	Over-weight	Obesity	Morbid Obesity
Headache	0.0%	18.8%	14.8%	16.7%	0.0%	25.0%	31.5%	22.2%	25.0%	0.0%
Muscle pain	0.0%	18.8%	14.8%	25.0%	0.0%	0.0%	20.8%	29.6%	41.7%	33.3%
Joint pain	0.0%	12.5%	18.5%	25.0%	0.0%	0.0%	18.8%	29.6%	33.3%	0.0%
Injection site pain	75.0%	72.9%	63.0%	83.3%	33.3%	100.0%	68.8%	44.4%	75.0%	33.3%
Fatigue	0.0%	37.5%	25.9%	66.7%	33.3%	25.0%	47.9%	63.0%	83.3%	66.7%
Fever	0.0%	6.3%	3.7%	8.3%	0.0%	50.0%	14.6%	25.9%	25.0%	33.3%
Injection site swelling	25.0%	8.3%	14.8%	16.7%	33.3%	50.0%	12.5%	11.1%	33.3%	0.0%
Nausea	0.0%	2.1%	3.7%	16.7%	0.0%	0.0%	12.5%	11.1%	0.0%	0.0%
Injection site redness	25.0%	6.3%	18.5%	16.7%	33.3%	0.0%	12.5%	7.4%	8.3%	0.0%
Lymphatic nodes enlargement	25.0%	6.3%	14.8%	16.7%	0.0%	25.0%	12.5%	14.8%	16.7%	0.0%
Insomnia	0.0%	4.2%	7.4%	0.0%	0.0%	0.0%	8.3%	11.1%	16.7%	0.0%
Limb pain	50.0%	54.2%	51.9%	75.0%	66.7%	25.0%	25.0%	37.0%	58.3%	66.7%
Injection site itching	25.0%	8.3%	3.7%	0.0%	0.0%	0.0%	2.1%	3.7%	8.3%	33.3%
Lethargy	0.0%	14.6%	11.1%	25.0%	0.0%	25.0%	16.7%	25.9%	25.0%	0.0%
Soreness	0.0%	2.1%	0.0%	0.0%	0.0%	0.0%	0.0%	7.4%	8.3%	0.0%
Other *	0.0%	0.0%	3.7%	8.3%	0.0%	0.0%	2.1%	3.7%	0.0%	0.0%

**Table 6 vaccines-10-00558-t006:** Two-sided *p*-values for difference in IgG counts between body mass index (BMI) groups. * The significant difference.

		Underweight	Normal Weight	Overweight	Obesity	Morbid Obesity
1st IgG testing	Underweight		0.3281	0.3377	0.8059	0.2221
	Normal weight			0.1245	0.3623	<0.0001 *
	Overweight				0.1494	0.0463 *
	Obesity					0.0388 *
	Morbid obesity					
2nd IgG testing	Underweight		0.8342	0.6117	0.7737	0.7472
	Normal weight			0.5136	0.3528	<0.0001 *
	Overweight				0.2057	0.9432
	Obesity					0.5839
	Morbid obesity					
3rd IgG testing	Underweight		0.3856	0.0180 *	0.3652	0.7798
	Normal weight			0.1000	0.6948	0.0198 *
	Overweight				0.5339	0.0199 *
	Obesity					0.3922
	Morbid obesity					

**Table 7 vaccines-10-00558-t007:** Two-sided *p*-values for difference in laboratory tests between body mass index (BMI) groups. * The significant difference. ** The two-tailed significance probability that could not be measured, therefore *p*-value for equal variance is assumed.

		Underweight	Normal Weight	Overweight	Obesity	Morbid Obesity
QuantiFERON Ag1 (CD4+)	Underweight		0.0378	0.1070	0.3158	0.3107
	Normal weight			0.2963	0.5932	0.0145 *
	Overweight				0.4275	0.0214 *
	Obesity					0.2830
	Morbid obesity					
QuantiFERON Ag2 (CD4+ & CD8+)	Underweight		0.0648	0.0852	0.2101	0.7831
	Normal weight			0.6542	0.4185	0.0538
	Overweight				0.3459	0.0681
	Obesity					0.2032
	Morbid obesity					
QuantiFERON Ag3	Underweight		0.4684	0.5208	0.5778	0.3921
	Normal weight			0.4374	0.3099	0.0368 *
	Overweight				0.6640	0.0161 *
	Obesity					0.1387
	Morbid obesity					
Overall IFN-γ >10	Underweight		0.0719 **	0.0566 **	0.1528 **	
	Normal weight			0.2877	0.5327	
	Overweight				0.8726	
	Obesity					
	Morbid obesity					

**Table 8 vaccines-10-00558-t008:** Cell-mediated immunity by body mass index (BMI) groups.

	Negative	Border Line	Positive	Time from 2nd Vaccination to QuantiFERON Collection (Days)
Underweight	75.0	0.0	25.0	184.3 ± 120.4
Normal weight	45.8	10.4	43.8	142.0 ± 94.0
Overweight	55.6	3.7	40.7	186.4 ± 108.9
Obesity	50.0	0.0	50.0	161.7 ± 110.8
Morbid obesity	66.7	0.0	33.3	84.7 ± 3.8

## Data Availability

The data presented in this study are available on request from the corresponding author.

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
