# Peer review of "Effectiveness, Adverse Events, and Immune Response Following Double Vaccination with BNT162b2 in Staff at the National Comprehensive Cancer Center (NCCC)"

_vaccines, 2022, doi:10.3390/vaccines10040558_

Round 1

Reviewer 1 Report

The manuscript entitled “Effectiveness, adverse events, and immune response following two double vaccination with BNT162b2 in workers at the National 3 Comprehensive Cancer Center (NCCC).”

The authors described the effectiveness, AE, and immune responses in the BNT162b2 vaccinee. The manuscript is straightforward, and however, there are some concerns.

Major comments:
1. The sample size is small, so there are brutal to conclude the data, e.g., the effect of obesity on AE and vaccine effectiveness.
2. The authors should label the y-axis in all figures present in the manuscript. Also, the figure legend should include more details, e.g., IgG count, Ag1, Ag2, Ag3, etc.
3. It will be more informative if the authors present the immune status (antibody titers, Interferon levels) in these breakthrough infection cases, especially from the before and after second vaccination groups.
4. I suggest representing the antibody titers in GMT (95%CI) and geometric mean fold change.

Author Response

Dear Editor,

I would like to resubmit our manuscript named “Effectiveness, adverse events, and immune response following double vaccination with BNT162b2 in staff at the National Comprehensive Cancer Center (NCCC)” (vaccines-1624772) to the journal for a publication. We would like to thank to a reviewer for his/her time and effort. We have followed all suggestions and tried to improve our manuscript accordingly. The revised manuscript (including changes suggested by all reviewers) is attached. Here is our response point by point.  

Q1. The sample size is small, so there are brutal to conclude the data, e.g., the effect of obesity on AE and vaccine effectiveness.

A1. Conclusions were revised. 

Q2. The authors should label the y-axis in all figures present in the manuscript. Also, the figure legend should include more details, e.g., IgG count, Ag1, Ag2, Ag3, etc.

A2. The y-axes were labelled in all Figures and more details were included in their legends.  

Q3. It will be more informative if the authors present the immune status (antibody titers, Interferon levels) in these breakthrough infection cases, especially from the before and after second vaccination groups.

A3. We appreciate this proposal however the stratification of vaccinated subjects was given by a protocol (COVID-SK001) and the adding of some other analyses would broaden manuscript in inappropriate manner we would like to avoid. 

Q4. I suggest representing the antibody titers in GMT (95%CI) and geometric mean fold change.

A4. IgG indexes and their 95% CI were added (see the section “2.4 IgG antibodies”). 

Reviewer 2 Report

Dear Editor,  I have now read the manuscript entiteled: “Effectiveness, adverse events, and immune response following double vaccination with BNT162b2 in workers at NCCC” (manuscript vaccines-1624772) by Palaka P et al. The manuscript presents collected data from a clinical SARS-CoV-2 vaccine study performed on 94 individuals who were given SARS-CoV-2 vaccine twice with approximately 1 month interval. The 94 study participants were divided into male and female participants and adverse events and vaccine immunogenicity was tested. Furthermore, the 94 individuals were further divided into five different groups based on their BMI (ranging from underweight to morbid obesity. Vaccine efficacy against severe disease was presented to be 96,7percent (since two individuals became SARS-CoV-2 PCR-positive after the second vaccine dose). Humoral immunity was estimated to be 97,87 percent positive after the second vaccine dose. Cell-mediated immunity was tested using three different QuantiFERON antigen tests and IFN-gamma overall creation >10 test.

The study is performed in a health care worker cohort, and data from these categories of participants is often of great interest. However, some more detailed description are needed to better understand the presented data.

Comments and questions:

Q1. In paragraph 2.4. IgG antibodies (page 7) IgG serological data are given at 1st and 2nd IgG count and then between 2nd and 3rd count. However, the authors did not report against which antigen the IgG was measured against. This information specifying the IgG SARS-CoV-2 antigen specificity should be given, otherwise the interpretation of the data will be difficult to understand.

Q2. Did the authors study the levels of SARS-CoV-2 neutralizing antibodies in their serum samples?

Q3. Figure 5. IgG antibodies illustrated in this figure suggest that Difference of IgG counts between gender is shown. Unfortunately, the figure legend did not reveal how many individuals were tested in each gender group? This information could be added to the figure text.

Q4. In Figure 6. IgG antibodies illustrated in this figure suggest that Difference of IgG counts between body max index (BMI) groups is shown. Unfortunately, the figure legend did not reveal how many individuals were tested in each BMI group?. The division in genders for each BMI group is not given?. This information could be added to the figure text

Q5. The cell-mediated immunity was studied using the QuantiFERON test consisting of three different assays. In QuantiFERON Ag1 (CD4+ cells were assayed, in QuantiFERON Ag2 both CD4+ and CD8+ and natural killer cells were studied (as indicated in Discussion page 11) and finally in QuantiFERON Ag3 CD4+, CD8+ and CD16+ (B- and T-cells) were studied.

So, which cell-types were significantly activated? Was it the CD4+, CD8+ the NK-cells or the B-cells that are presented in Figure 7 (QuantiFERON by gender) ? and in Figure 8 (QuantiFERON results by body mass index (BMI) groups. ?

Q6. Figure 8 is very squeezed and small on page 9. This figure 8 should be shown in a clearer size and way.

Q7. In Figure 7 and in Table 8, the Interferon gamma overall creation is shown. The method for this analysis is lacking in the Materials and methods section. This method should be added to the Materials and methods section.

Q8. In Table 9 the Cell immunity by body mass index (BMI) groups is presented. Interestingly, the time for QuantiFERON sample collection and comparison seem to vary between the study groups.

Four of the BMI study groups in Table 9 were tested at a median day of 142 to 186,44 days after the second vaccination, but the morbid obesity group was tested at 84,67 days post 2nd vaccination? These mean day differences between study groups are difficult to understand. The authors should explain why different days were used, and how this may affect the results shown.

 Q9. In the Discussion page 10 the authors discuss and present the two cases that were shown to have become SARS-CoV-2 PCR-positive after the 2nd vaccination.

  1. Did these two individuals develop neutralizing serum antibodies to the SARS-CoV-2 virus variant that they became infected with?
  2. Which SARS-CoV-2 neutralization assay was performed?
  3. Since PCR was performed, perhaps the variant of SARS-CoV-2 was analyzed (was it beta, delta or omnicron ?).
  4. What was the SARS-CoV-2 virus variant used in the vaccine?
  5. How did these two individuals respond in the QuantiFERON assays?

Author Response

Dear Editor,

I would like to resubmit our manuscript named “Effectiveness, adverse events, and immune response following double vaccination with BNT162b2 in staff at the National Comprehensive Cancer Center (NCCC)” (vaccines-1624772) to the journal for a publication. We would like to thank to a reviewer for his/her time and effort. We have followed all suggestions and tried to improve our manuscript accordingly. The revised manuscript (including changes suggested by all reviewers) is attached. Here is our response point by point.

Q1. In paragraph 2.4. IgG antibodies (page 7) IgG serological data are given at 1st and 2nd IgG count and then between 2nd and 3rd count. However, the authors did not report against which antigen the IgG was measured against. This information specifying the IgG SARS-CoV-2 antigen specificity should be given, otherwise the interpretation of the data will be difficult to understand.

A1. This was clarified in the section “4.3 IgG antibody measurement”.  

Q2. Did the authors study the levels of SARS-CoV-2 neutralizing antibodies in their serum samples?

A2. This was also clarified in the section “4.3 IgG antibody measurement”.  

Q3. Figure 5. IgG antibodies illustrated in this figure suggest that Difference of IgG counts between gender is shown. Unfortunately, the figure legend did not reveal how many individuals were tested in each gender group? This information could be added to the figure text.

A3. This information was added.

Q4. In Figure 6. IgG antibodies illustrated in this figure suggest that Difference of IgG counts between body max index (BMI) groups is shown. Unfortunately, the figure legend did not reveal how many individuals were tested in each BMI group. The division in genders for each BMI group is not given. This information could be added to the figure text.

A4. This information was added. 

Q5. The cell-mediated immunity was studied using the QuantiFERON test consisting of three different assays. In QuantiFERON Ag1 (CD4+ cells were assayed, in QuantiFERON Ag2 both CD4+ and CD8+ and natural killer cells were studied (as indicated in Discussion page 11) and finally in QuantiFERON Ag3 CD4+, CD8+ and CD16+ (B- and T-cells) were studied. So, which cell-types were significantly activated? Was it the CD4+, CD8+ the NK-cells or the B-cells that are presented in Figure 7 (QuantiFERON by gender) and in Figure 8 (QuantiFERON results by body mass index (BMI) groups?

A5. This was explained below Figures 7 and 8 and in the Discussion.

Q6. Figure 8 is very squeezed and small on page 9. This figure 8 should be shown in a clearer size and way.

A6. Figure 8 was modified.

Q7. In Figure 7 and in Table 8, the Interferon gamma overall creation is shown. The method for this analysis is lacking in the Materials and methods section. This method should be added to the Materials and methods section.

A7. The method was added.

Q8. In Table 9 the Cell immunity by body mass index (BMI) groups is presented. Interestingly, the time for QuantiFERON sample collection and comparison seem to vary between the study groups. Four of the BMI study groups in Table 9 were tested at a median day of 142 to 186,44 days after the second vaccination, but the morbid obesity group was tested at 84,67 days post 2nd vaccination? These mean day differences between study groups are difficult to understand. The authors should explain why different days were used, and how this may affect the results shown.

A8. An explanation was added (see the Discussion).

 Q9. In the Discussion page 10 the authors discuss and present the two cases that were shown to have become SARS-CoV-2 PCR-positive after the 2nd vaccination.

  1. Did these two individuals develop neutralizing serum antibodies to the SARS-CoV-2 virus variant that they became infected with?
  2. Which SARS-CoV-2 neutralization assay was performed?
  3. Since PCR was performed, perhaps the variant of SARS-CoV-2 was analyzed (beta, delta or omicron?)
  4. What was the SARS-CoV-2 virus variant used in the vaccine?
  5. How did these two individuals respond in the QuantiFERON assays?

A9. All available data were added (see the Discussion).

Reviewer 3 Report

 The authors designed a study to partially measure the antibody and cell immunity response after vaccination. They conducted a 6-month prospective study involving workers of NCCC in Slovakia, who were tested for IgG antibody and cell immune responses after double vaccination with BNT162b2. IgG antibodies were detected at 3, 7, and 26 weeks, respectively. At 6 months, blood samples were tested by two different Interferon-γ  release assays to determine responses to Spike protein antigen and Nucleocapsid protein antigen of  the novel coronavirus. Results were stratified by gender and body mass index (BMI). Statistical significance was set at p=0.05.

The article deals with an important health aspect of the COVID-19 pandemic and gives specific results. The sample is small so authors need to mention this in the Discussion as part of the limitations of the study.

For all the test add Tables to show which values were greater. For instance if the difference of antibodies  between underweight and obese is significant then show which is greater. This can be observed with the bar graphs but it is better to have a summary table.

Split Figure 8. It cannot be seen very well.

The methodology to determine the vaccine efficacy should be compared with other works. A summary Table highlighting differences would help.

In the Discussion it is necessary to mention all the limitations of this study.

In the Discussion please emphasize in what directions are the significant differences between groups. For instance, efficacy was lower for obese people. This should be done for all the significant differences.

The Methods section usually is before Discussions. Not sure if there is a particular reason to leave the Methods Section at the end.

The discussion section needs to include more comparison with other similar studies regarding all the variables that have been studied in this manuscript. Please also add some discussion of potential cofounders as part of the statistical results.

Finally discuss more broadly the potential/factual implications of these results.

Author Response

Dear Editor,

I would like to resubmit our manuscript named “Effectiveness, adverse events, and immune response following double vaccination with BNT162b2 in staff at the National Comprehensive Cancer Center (NCCC)” (vaccines-1624772) to the journal for a publication. We would like to thank to a reviewer for his/her time and effort. We have followed all suggestions and tried to improve our manuscript accordingly. The revised manuscript (including changes suggested by all reviewers) is attached. Here is our response point by point.

Q1. The sample is small so authors need to mention this in the Discussion as part of the limitations of the study.

A1. The limitations of this study including small number of subjects were added to the Discussion.  

Q2. For all the tests add Tables to show which values were greater. For instance, if the difference of antibodies between underweight and obese is significant then show which is greater. This can be observed with the bar graphs but it is better to have a summary table.

A2. Since another reviewer suggested to omit some Tables due to their high number, we decided to add values into the text (see the section “2.4 IgG antibodies”).    

Q3. Split Figure 8. It cannot be seen very well.

A3. Figure 8 was split.

Q4. The methodology to determine the vaccine efficacy should be compared with other works. A summary Table highlighting differences would help.

A4. Since another reviewer suggested to omit some Tables, we decided to compare the vaccine efficacy with some other studies within the Discussion, not in added Table.      

Q5. In the Discussion it is necessary to mention all the limitations of this study.

A5. The limitations of this study were mentioned in the Discussion.

Q6. In the discussion, please emphasize in what directions are the significant differences between groups. For instance, efficacy was lower for obese people. This should be done for all the significant differences.

A6. The discussion was revised.

Q7. The Methods section usually is before Discussions. Not sure if there is a particular reason to leave the Methods Section at the end.

A7. The structure of a manuscript is given by the Vaccines Journal.

Q8. The discussion section needs to include more comparison with other similar studies regarding all the variables that have been studied in this manuscript. Please also add some discussion of potential cofounders as part of the statistical results.

A8. The discussion was revised.

Q9. Finally discuss more broadly the potential/factual implications of these results.

A9. The discussion was revised.

Reviewer 4 Report

This article is about a prospective observational study looking at the effect of double vaccination with BNT162b2, which is a vaccine against SARS-CoV-2. While it provides valuable data, the quality of the manuscript is not high enough for a scientific paper. The following points require improvement.

1. I think there are unnecessarily many Tables and Figures. For example, Table 2 should be switched to a supplementary table. Also, wouldn't it be possible to change Figure 6 into a line chart, and to indicate in the Figure whether there is a significant difference or not? By doing so, it is possible to remove one excessive table (Table 7).

2. I feel that the numbers listed in the Table are too detailed. For example, would it be possible to limit the data in Table 6 to one decimal place?

3. The authors used categories of underweighted (Ë‚18.5 85 kg/m2), normal weight (18.5-24.9 kg/m2), overweight (25.0-29.9 kg/m2), obese (30.0-34.9 kg/m2), morbid obesity (≥35.0 kg/m2). Is this division common in scientific papers? Could you cite the appropriate literature?

4. In Table 6, the data after the second vaccination for morbid obesity appears to be missing.

5. Did any of the subjects require a leave of absence due to fever or fatigue associated with the vaccination? The number of days required for leave would be very valuable data.

6. I would think you need to mention the limitations of this study in the Discussion. For example, I think it is a major limitation that only 3 subjects were classified as morbid obesity when categorized by weight.

Author Response

Dear Editor,

I would like to resubmit our manuscript named “Effectiveness, adverse events, and immune response following double vaccination with BNT162b2 in staff at the National Comprehensive Cancer Center (NCCC)” (vaccines-1624772) to the journal for a publication. We would like to thank to a reviewer for his/her time and effort. We have followed all suggestions and tried to improve our manuscript accordingly. The revised manuscript (including changes suggested by all reviewers) is attached. Here is our response point by point.

Q1. I think there are unnecessarily many Tables and Figures. For example, Table 2 should be switched to a supplementary table. Also, wouldn't it be possible to change Figure 6 into a line chart, and to indicate in the Figure whether there is a significant difference or not? By doing so, it is possible to remove one excessive table (Table 7).

A1. Table 2 was removed. Since we changed the other Tables, Figures and their Legends according to the reviewers, we would like to keep Table 7 in a manuscript.

Q2. I feel that the numbers listed in the Table are too detailed. For example, would it be possible to limit the data in Table 6 to one decimal place?

A2. The data in Table 6 were limited to one decimal place.

Q3. The authors used categories of underweighted (Ë‚18.5 85 kg/m2), normal weight (18.5-24.9 kg/m2), overweight (25.0-29.9 kg/m2), obese (30.0-34.9 kg/m2), morbid obesity (≥35.0 kg/m2). Is this division common in scientific papers? Could you cite the appropriate literature?

A3. We used BMI categories by the Centers for Disease Control and Prevention. An appropriate reference was added.

Q4. In Table 6, the data after the second vaccination for morbid obesity appears to be missing.

A4. The missing data were added to Table 6.

Q5. Did any of the subjects require a leave of absence due to fever or fatigue associated with the vaccination? The number of days required for leave would be very valuable data.

A5. No subject within this study required a leave of absence due to fever or fatigue associated with the vaccination. This information was added to the Discussion.

Q6. I would think you need to mention the limitations of this study in the Discussion. For example, I think it is a major limitation that only 3 subjects were classified as morbid obesity when categorized by weight.

A6. This study limitations were added to the Discussion.

Round 2

Reviewer 1 Report

The authors addressed my concerns so I no longer have further questions/comments.

Author Response

Thanks for your valuable comments.

Reviewer 3 Report

The authors have improved the paper. The authors mentioned the limitations due to small samples.

Author Response

Thanks for your valuable comments.

Reviewer 4 Report

The authors have revised their original manuscript according to the reviewers’ comments. I would think that this revised manuscript is better organized and suitable for publication in journal.

Author Response

Thanks for your valuable comments.